

# Comparison of serum exosome isolation methods on co-precipitated free microRNAs

Yirui Cheng[1], Xiangyun Qu[2], Zhaonan Dong[2], Qingyu Zeng[1], Xueqing Ma[2], Yunli Jia[2], Ruochen Li[2], Xiaoxu Jiang[2], Cecilia Williams[3], Tao Wang[2] and Weiliang Xia[1]

[1] School of Biomedical Engineering and Med-X Research Institute, Shanghai Jiao Tong University, Shanghai, China
[2] Jiangsu Cancer Molecular Diagnostics Engineering Research Center, Suzhou MicroDiag Biomedicine Co., Ltd, Suzhou, China
[3] Department of Protein Science, KTH Royal Institute of Technology, Science for Life Laboratory, Solna, Sweden

Corresponding author
Weiliang Xia, wlxia@sjtu.edu.cn

## ABSTRACT

**Background**. Exosomes are nano-sized extracellular vesicles containing different biomolecules such as proteins and microRNAs (miRNAs) that mediate intercellular communication. Recently, numerous studies have reported the important functions of exosomal miRNAs in disease development and the potential clinical application as diagnostic biomarkers. Up to now, the most commonly used methods to extract exosomes are ultracentrifugation (UC) and precipitation-based commercial kit (e.g., ExoQuick). Generally, both UC and ExoQuick method could co-isolate contaminating proteins along with exosomes, with the UC method yielding even purer exosomes than ExoQuick. However, the comparison of these two methods on co-precipitated free miRNAs is still unknown.

**Methods**. In this study, we isolated exosomes from the human serum with exogenously added cel-miR-39 by UC and ExoQuick and compared the proportion of cel-miR-39 co-precipitated with exosomes extracted by these two methods.

**Results**. Using exogenous cel-miR-39 as free miRNAs in serum, we concluded that ExoQuick co-isolates a small proportion of free miRNAs while UC hardly precipitates any free miRNAs. We also found that incubation at 37 °C for 1 h could decrease the proportion of free miRNAs, and exosomal miRNAs like miR-126 and miR-152 also decreased when RNase A was used. In conclusion, our findings provide essential information about the details of serum exosome isolation methods for further research on exosomal miRNAs.

# INTRODUCTION

Exosomes are small secreted extracellular vesicles of 30–200 nm in diameter with the same structure as cell membrane (*Pegtel & Gould, 2019*; *Thery, Zitvogel & Amigorena, 2002*; *Tkach & Thery, 2016*). They convey intercellular communications by delivery of biomolecules and affect multiple physiological processes under normal or diseased

conditions (*He et al., 2018*; *Meckes Jr, RH & Raab-Traub, 2010*; *Valadi et al., 2007*). They originate from multivesicular bodies (MVBs) which contain many small vesicles called intraluminal endosomal vesicles (ILVs) and the ILVs become exosomes when the MVBs fuse with the plasma membrane, releasing the ILVs into the extracellular space (*Colombo, Raposo & Thery, 2014*). Moreover, exosomes have been found in diverse biological fluids, such as serum, urine, saliva and breast milk (*Thery et al., 2006*).

MicroRNAs (miRNAs), a class of 19–23 nt non-coding RNAs are known as a mediator of post-transcriptional regulation, which can negatively regulate the expression of target mRNAs (*Bartel, 2004*). While the majority of miRNAs are located within the cell, recently, significant number of miRNAs have been found in extracellular environment, including various biological fluids and cell culture media, commonly known as circulating miRNAs or extracellular miRNAs (*Turchinovich et al., 2011*). Recently, an increasing number of extracellular miRNAs have been detected in exosomes isolated from biological fluids and cell culture media (*Valadi et al., 2007*). Moreover, some exosomal miRNAs have been shown to regulate disease development and reported as biomarkers of different types of cancers, cardiovascular disease and brain injury (*Grimolizzi et al., 2017*; *Lugli et al., 2015*; *Zhang et al., 2017*). In addition to exosomal miRNAs, there are still some circulating miRNAs free in the serum (*Mitchell et al., 2008*). It remains unclear whether these free miRNAs can be co-precipitated with exosomes and how to remove these free miRNAs from exosome pellets if so.

The most commonly used methods to extract exosomes are ultracentrifugation (UC) and commercially available kits, like ExoQuick from Systems Biosciences, exploiting sedimentation, with pros and cons for each method (*Alvarez et al., 2012*; *Umezu et al., 2013*). Generally, UC is the most reliable "gold standard" method but time-consuming; and precipitation methods such as ExoQuick can obtain higher yields of exosomes but co-precipitate more impure proteins at the same time (*Shao et al., 2016*). However, if and to what extent these two exosome isolation methods can co-isolate serum free miRNAs remain to be investigated.

In this study, regarding exogenous cel-miR-39 as free miRNAs in serum, we concluded that ExoQuick co-isolates a small proportion of free miRNAs while UC hardly precipitates any free miRNAs. Further, we found that incubation at 37 °C for 1 h can decrease the proportion of free miRNAs in exosome pellets and exosomal miRNAs like miR-126 and miR-152 (*Grimolizzi et al., 2017*; *Lugli et al., 2015*; *Ong et al., 2014*; *Zhang et al., 2017*) also decreased when RNase A was used. In conclusion, our findings provide essential information on the details of serum exosome isolation methods for further studies on exosomal miRNAs.

## MATERIALS & METHODS

### Sample collection

Human blood samples were collected into serum collection tubes from the Shanghai Chest Hospital. Blood samples were centrifuged at 3000 rpm at 4 °C for 10 min for serum collection, followed by an additional centrifugation step at 3,000 g for 15 min.

The supernatants of different samples were mixed and then equally divided into several portions (500 μL) for subsequent experiments.

This study was approved by the Ethical Committee of the School of Biomedical Engineering, Shanghai Jiao Tong University (Registration No. 2019045) and carried out in accordance with the Declaration of Helsinki. All participants provided written informed consent.

## Exogenous miRNA addition

The serum (500 μL) were added with 5 μL $7\times10^9$ copies/μL cel-miR-39 or 5 μL $7\times10^6$ copies/μL cel-miR-39 or 5 μL ddH$_2$O. The sequence of cel-miR-39 is: forward 5′-UCACCGGGUGUAAAUCAGCUUG-3′.

## Exosome isolation

### ExoQuick$^{TM}$ Kit

Exosomes from 500 μL serum were isolated by ExoQuick$^{TM}$ kit (System Biosciences Inc., CA, USA) according to the manufacturer's recommendations. Briefly, 120 μL EXOQ5A-1 was added to serum and the mixture was incubated at 4 °C for 30 min. After incubation, the mixture was centrifuged at 13,000 rpm for 2 min. The supernatant was removed and the exosome pellet was used for subsequent experiments.

### Ultracentrifugation (UC)

Exosomes from 500 μL serum were also isolated by UC method. Briefly, the serum was transferred into 12.5 mL ultracentrifuge tube (Beckman). The tube then was filled with PBS (HyClone, SH30256.01) followed by ultracentrifugation twice for 70 min at 100,000 g, 4 °C in an SW 40Ti swinging-bucket rotor (Beckman). The supernatant was discarded and the exosome pellet was used for subsequent experiments.

## Exosome identification

### Transmission electron microscopy (TEM)

The exosome pellet was resuspended in a small volume of PBS. According to Thery et al. (*Thery et al., 2006*), the exosomes (5 μL) was dripped onto a copper grid (Zhongjingkeyi, CHN, BZ110223b) firstly. After one minute, the droplet was sucked out using the air-laid paper. And then, the 2% uranyl acetate (Merck, 1005) solution (5 μL) was dripped onto the same copper grid for negative-staining and sucked out again one minute later.

### Nanoparticle tracking analysis (NTA)

The size distribution of the exosome pellet was measured by ZetaView (Particle Metrix). The pellet was resuspended in a proper volume of PBS to achieve the optimal detectable concentration (about $10^7$ particles per mL) of its corresponding software (ZetaView 8.03.04.01). For each measurement, 3–5 mL of the diluted sample was injected into the machine.

### Western blot

Western blotting was performed according to the manufacturer's instruction manual. Briefly, each pellet (30 μg) was mixed with loading buffer (5x) and heated at 95 °C for 5

min. Proteins were loaded on 10% sodium dodecyl sulfate polyacrylamide gel (SDS-PAGE; EpiZyme, PG112) and separated at constant 55 V for 30 min first and then 120 V until the loading buffer running out of the SDS-PAGE. Then, proteins were transferred to a nitrocellulose membrane (GE Healthcare, 10600002) at constant 300 mA for 1.5 h. Next, the membrane was blocked with 5% nonfat milk powder suspended in tris-buffered saline and tween 20 (TBST) for 1 h at room temperature. The blots were probed with TSG101 (Abcam, 133586) and subsequently probed with horseradish peroxidase conjugated secondary anti-rabbit antibodies (Jackson). Finally, the blots were visualized using the enhanced chemiluminescent (ECL; Thermo, 1856136) and chemiluminescent imaging system (Tanon, 5200).

## RNase A and RNase Inhibitor treatment

The exosome pellet was suspended in 500 µL PBS and a determined volume of RNase A (Thermo Scientific, MAN0012003) was added in the RNase A treatment groups to reach a final concentration of 10 µg/mL, 20 µg/mL or 30 µg/mL. And RNase inhibitor (Thermo Scientific, EO0381) was added in the RNase inhibitor treatment groups to reach a final concentration of 1 U/µL. Then the mixture was incubated at 37 °C for 1 h.

## RNA isolation

Exosomal RNA was extracted by miRNeasy Mini Kit (Qiagen, 217004) according to the manufacturer's recommendations. The final elution volume was 35 µL.

## Reverse Transcription Polymerase Chain Reaction (RT-PCR)

The cDNA of cel-miR-39, miR-126 and miR-152 were synthesized with the Revert Aid First Strand cDNA Synthesis Kit (Thermo Scientific, K1622). Briefly, 5 µL total RNA from each sample were mixed with 5 µL of RT-Primer, 2 µL of 10×T4 Buffer, 2 µL of 10mM dNTP Mix, 1 µL of RNase inhibitor, 1 µL of Transcriptase, 0.5 µL of T4 Polynucleotide Kinase, 0.25 µL of T4 Kinase, and 3.25 µL of nuclease-free water. Then the 20 µL of mixture was incubated at 16 °C for 30 min, 42 °C for 30 min, 85 °C for 5 min followed by cooling at 4 °C in the PCR instrument (LongGene, A100).

Note that 5 µL of 35 µL extracted RNA were used in RT-PCR, so if all the exogenous cel-miR-39 had been co-isolated with exosomes, the current concentration of cel-miR-39 in this procedure would be $10^9$ copies/µL or $10^6$ copies/µL. Therefore, $10^9$ copies/µL or $10^6$ copies/µL cel-miR-39 were used as a standard control.

Here, stemloop RT primers were used for cDNA synthesis. The sequence of miR-126 is: 5′-UCGUACCGUGAGUAAUAAUGCG-3′. The RT primer of miR-126 is: 5′-GATGAGGAGTGTCGTGGAGTCGGCAATTTCCTCATCACGCATTA-3′. The sequence of miR-152 is: 5′-AGGUUCUGUGAUACACUCCGACU-3′. The RT primer of miR-152 is: 5′- GATGAGGAGTGTCGTGGAGTCGGCAATTTCCTCATCAAGTCGGAG-3′. The RT primer of cel-miR-39 is: 5′- GATGAGGAGTGTCGTGGAGTCGGCAATTTCCTCATC-CAAGCTG -3′.

## Quantitative real-time PCR (qRT-PCR)

To validate the quantity of cel-miR-39, miR-126 and miR-152 in the RNA samples, qPCR was performed using Fluorescence Polymerase Chain Reaction (PCR) Detection

Kit for the Analysis of Human miRNA Gene Expression (Jiangsu MicroDiag Biomedicine Technology Co. Ltd., China) according to the manufacturer's recommendations. Briefly, the synthesized cDNA of cel-miR-39, miR-126 and miR-152 from each sample were mixed with 10× Buffer, 25 mM MgCl2, 25 mM dNTP Mix, F-primer, R-primer, miRNA-specific probe, HS Tag, UDG and ddH2O. Meanwhile, $10^2$ - $10^7$ copies of cel-miR-39, miR-126 and miR-152 were amplified in this procedure to get a curve for calculating miRNA copy numbers in each sample.

The PCR reaction mixture was incubated at 37 °C for 5 min, and then 94 °C for 5 min, followed by 50 cycles of 94 °C for 15 s, 60 °C for 60 s, and cooling at 50 °C for 30 s, which was performed using a real-time fluorescent quantitative PCR instrument (Roche, LightCycler 480).

Here, the miRNA-specific probes were used. The F-primer of miR-126 is: 5′-CTCCGTCAGGGTCGTACCGTGAGTAA-3′. The R-primer of miR-126 is: 5′-CTCAAGTGTCGTGGAGTCGGC-3′. The specific probe of miR-126 is: 5′-FAM-TTTCCTCATCACGCATTA-MGB-3′. The F-primer of miR-152 is: 5′-CTCCGTCAGGGAGGTTCTGTGATACA-3′. The R-primer of miR-152 is: 5′-CTCAAGTGTCGTGGAGTCGGC-3′. The specific probe of miR-152 is: 5′-FAM-TTTCCTCATCAAGTCGGAG-MGB-3′. The F-primer of cel-miR-39 is: 5′-CGTATGAGCGTCACCGGGTGTAAATCA-3′. The R-primer of miR-152 is: 5′-CTCAAGTGTCGTGGAGTCGGCAA-3′. The specific probe of miR-152 is: 5′-FAM-TTTCCTCATCCAAGCTG-MGB-3′.

## Statistical analysis

Each experiment was repeated at least three times. Data were analyzed and all statistical graphs were generated by GraphPad Prism 6.0 (GraphPad Software Inc, La Jolla, CA, USA). Differences between two groups were analyzed using one-way or two-way ANOVA and the probability value below 0.05 was considered significant.

# RESULTS

## Workflow of this research and the identification of exosomes

The experimental procedure of this research is summarized (Fig. 1), with details given in subsequent sections. First, 5 µL $7 \times 10^9$ copies/µL or $7 \times 10^6$ copies/µL exogenous cel-miR-39 was added in 500 µL human serum while same volume of ddH$_2$O was added in another group as blank control (step I). Next, the exosomes were isolated by ExoQuick or UC method (step II) and characterized using TEM, NTA and western blot (Figs. 2A–2E). TEM showed exosomes' "cup-shaped" morphology and co-isolated impure proteins in both methods (Figs. 2A, 2D). NTA analysis showed exosomes' similar particle size distribution mostly between 30-200 nm (Figs. 2B, 2E), however, ExoQuick-isolated exosomes tended to be a little smaller than those from UC, which may be caused by more co-precipitated impure proteins from ExoQuick method. Furthermore, exosomes from both methods obtained the exosome-associated protein marker, CD63, TSG101, CD9 (Figs. 2C, 2F) (Thery et al., 2018). Following, to clear up the co-isolation free miRNA in exosome pellets, 10 µg/mL, 20 µg/mL or 30 µg/mL RNase A were added in RNase A group and incubated

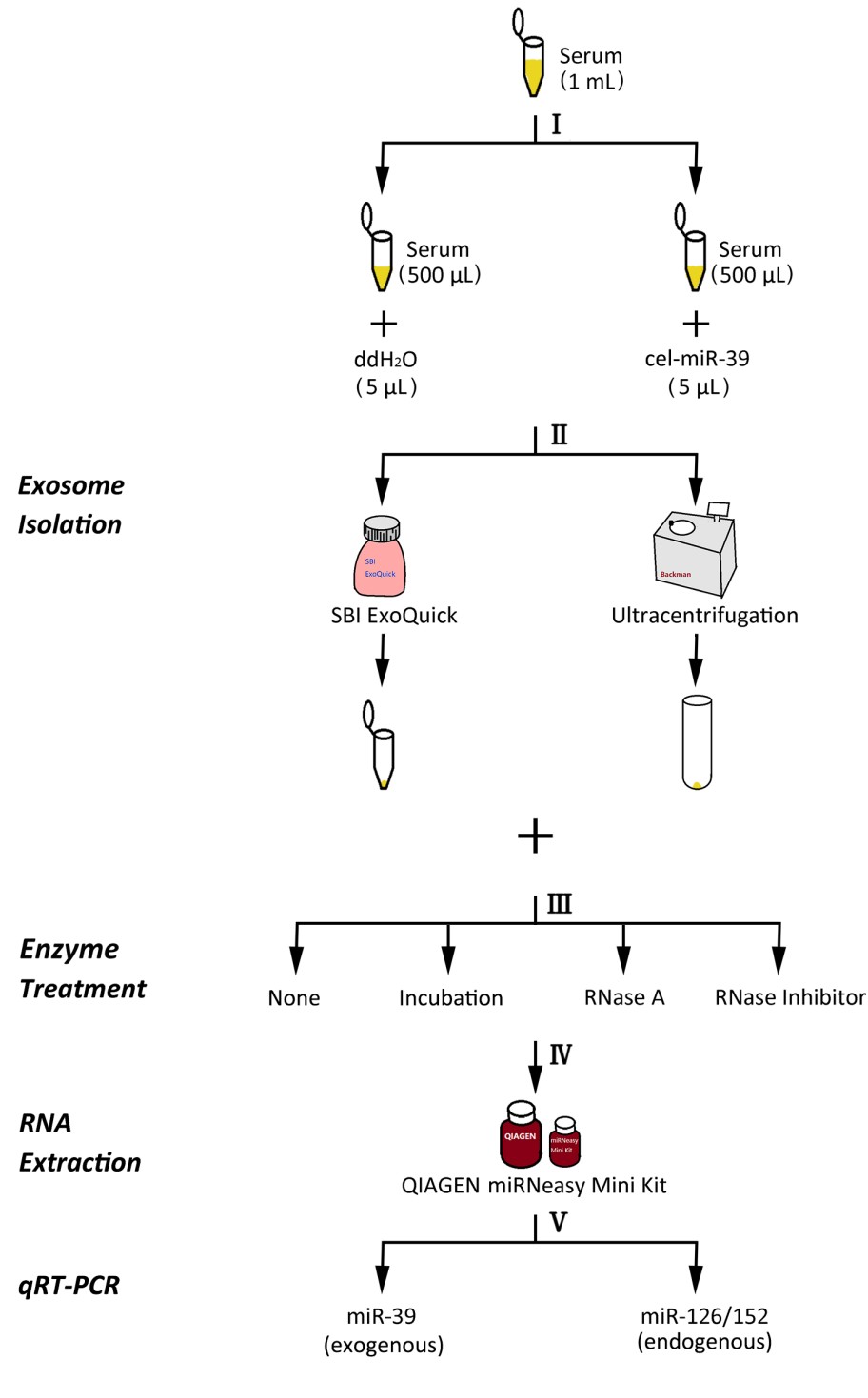

**Figure 1** **The experimental flowchart of this research.** (I) exogenously added cel-miR-39 in serum, blank control: ddH$_2$O, (II) isolation of exosomes by ExoQuick or UC method, (III) using 10 $\mu$g/mL, 20 $\mu$g/mL, 30 $\mu$g/mL RNase A or 1 U/$\mu$L RNase inhibitor or nothing with incubation at 37 °C for 1 h or nothing (control) to treat exosome pellets, (IV) extraction of total RNA including miRNA by miRNeasy Mini Kit, (V) detecting the quantity of cel-miR-39, miR-126 or miR-152 in exosome samples by qRT-PCR.

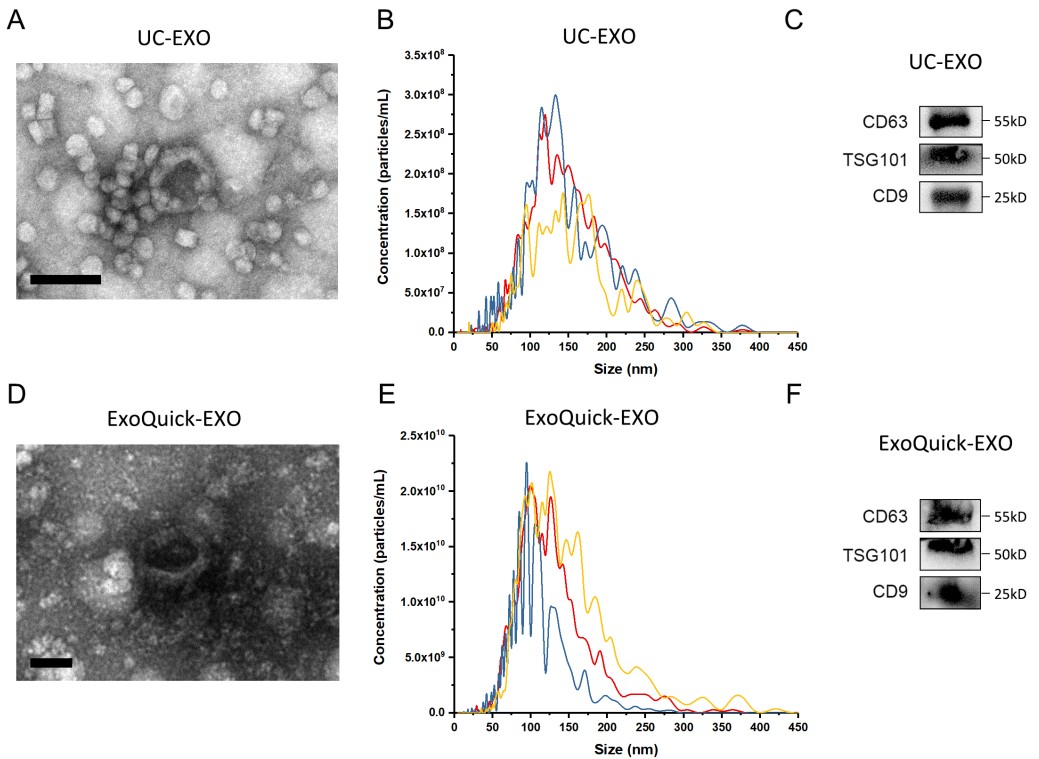

**Figure 2** **The characterization of exosomes isolated from human serum.** (A, D) The morphology of exosomes. Scale bars: 200 nm. (B, E) The particle size distribution of exosomes. (C, F) The level of exosome-associated protein, CD63, TSG101 and CD9.

with exosomes at 37 °C for 1 h. Moreover, 1 U/μL RNase inhibitor were used to eliminate the effect of environmental RNase; the control group was a blank control with no enzyme addition, which was also placed at 4 °C for 1 h (step III). Then, total RNA including miRNA of exosomes was extracted by QIAGEN miRNeasy Mini Kit and cel-miR-39, miR-126 or miR-152 in exosome samples were detected using RT-PCR followed by qPCR. 5 μL $10^9$ copies/μL or $10^6$ copies/μL cel-miR-39 was used as standard control in PCR procedure (step IV, V and VI).

## ExoQuick co-isolates a small proportion of free miRNAs

As described in Fig. 1, higher concentration ($7 \times 10^9$ copies/μL) of cel-miR-39 was added in 500 μL serum and then exosomes were extracted by ExoQuick method. The ratio of cel-miR-39 in exosome samples to standard cel-miR-39 sample ($10^9$ copies/μL) showed that about 2% exosome-independent cel-miR-39 were co-isolated with exosomes extracted by ExoQuick method (Fig. 3A). To measure the effect of RNase A to remove exosome-independent miRNAs, 10 μg/mL, 20 μg/mL, 30 μg/mL RNase A and 1 U/μL RNase inhibitor were respectively added in the same exosome samples, and the results of PCR showed that RNase A can decrease the quantity of co-isolated cel-miR-39 in exosomes but there are no significance difference between different concentrations of RNase A and RNase inhibitor groups (Fig. 3A). Moreover, RNase A decreased the quantity of exosomal

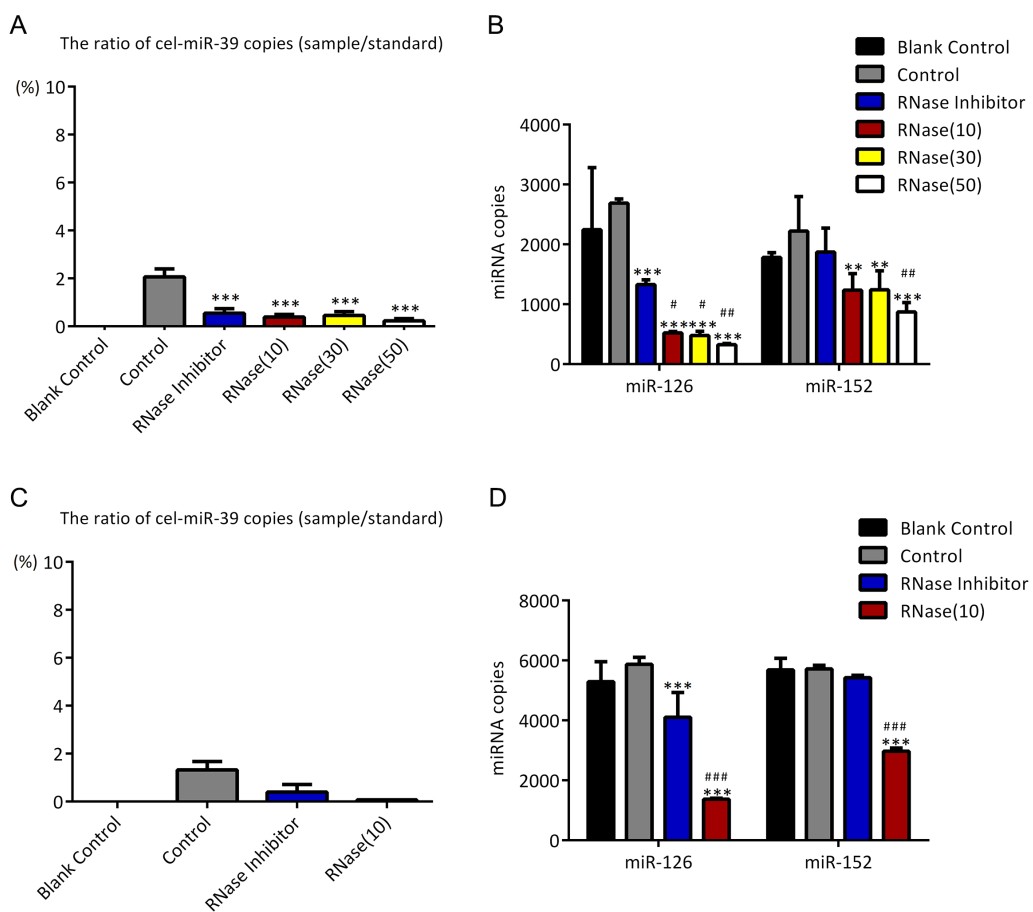

**Figure 3** **ExoQuick method co-isolates a small proportion of free miRNAs.** (A–B) A high concentration ($7 \times 10^9$ copies/$\mu$L) of cel-miR-39 was added in serum. Exosomes were extracted by the ExoQuick method. (A) The ratio of co-precipitated cel-miR-39 in different samples to standard sample ($10^9$ copies/$\mu$L cel-miR-39). ***$p < 0.001$ vs. Control. (b) The amount of exosomal miR-126 and miR-152 in different samples. **$p < 0.01$, ***$p < 0.001$ vs. Control. #$p < 0.05$, ##$p < 0.01$ vs. RNase Inhibitor. (C–D) Low concentration ($7 \times 10^6$ copies/$\mu$L) of cel-miR-39 were added in serum. Exosomes were extracted by the ExoQuick method. (C) The ratio of co-precipitated cel-miR-39 in different samples to standard sample ($10^6$ copies/$\mu$L cel-miR-39). (D) The amount of exosomal miR-126 and miR-152 in different samples. ***$p < 0.001$ vs. Control. ###$p < 0.001$ vs. RNase Inhibitor.

miR-126 and miR-152 at the same time compared with control and RNase inhibitor groups (Fig. 3B). Interestingly, RNase inhibitor reduced the amount of exosomal miR-126 but not reduced miR-152 (Fig. 3B).

To verify if the co-isolation is due to the high concentration of cel-miR-39, low concentration ($7\times10^6$ copies/$\mu$L) of miR-39 were exogenously added in serum. Similar to the results of high concentration cel-miR-39 experiment, nearly 2% cel-miR-39 were co-isolated with exosomes and RNase A decreased the amount of exosomal miR-126 and miR-152 (Figs. 3C–3D). And RNase inhibitor reduced the amount of exosomal miR-126, not the miR-152 in low concentration cel-miR-39 experiment as well (Fig. 3D). These results indicated that ExoQuick method can co-isolate 2% free miRNA in serum

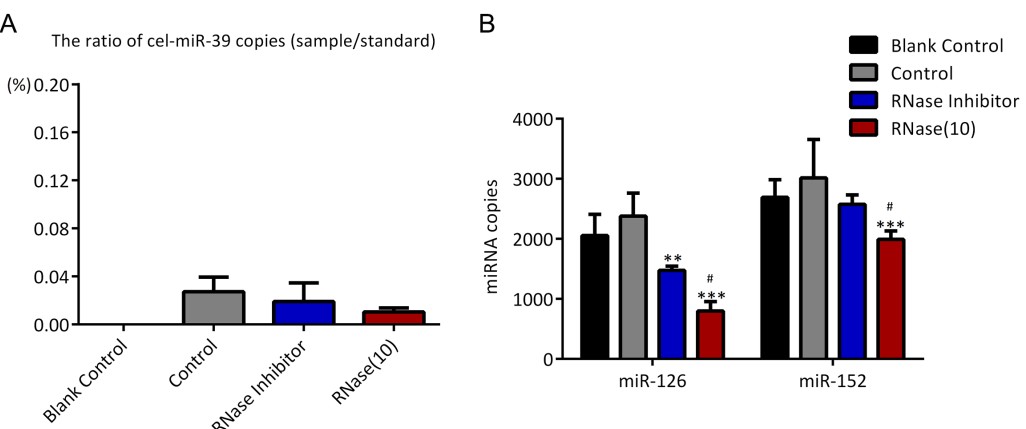

**Figure 4  Ultracentrifugation (UC) method co-isolates almost no proportion of free miRNAs.** (A-B) A high concentration ($7 \times 10^9$ copies/µL) of cel-miR-39 was added in serum. Exosomes were extracted by the UC method. (A) The ratio of co-precipitated cel-miR-39 in different samples to standard sample ($10^9$ copies/µL cel-miR-39). (b) The amount of exosomal miR-126 and miR-152 in different samples. ** $p <$ 0.01, *** $p < 0.001$ vs. Control. # $p < 0.05$ vs. RNase Inhibitor.

regardless of the concentrations of miRNAs and RNase A not only can reduce the number of exosome-independent miRNAs, but also reduce the miRNAs in the exosomes.

## Ultracentrifugation co-isolates almost no proportion of free miRNAs

To investigate if UC can co-isolate free miRNAs in serum with exosomes as ExoQuick method did, higher concentration ($7 \times 10^9$ copies/µL) of cel-miR-39 was added in 500 µL serum and then exosomes were extracted by UC method. The results showed that only 0.03% exogenous cel-miR-39 were co-isolated with exosomes extracted by UC method (Fig. 4A). Meanwhile, exosomal miR-126 and miR-152 were decreased by RNase A and miR-126 rather than miR-152 was reduced by RNase inhibitor (Fig. 4B). These results indicated that UC method hardly precipitate any exosome-independent miRNAs with exosomes and RNase A can reduce the miRNAs in exosomes no matter which method is used.

## Incubation at 37 °C for 1 h reduces the amount of co-isolated free miRNAs

In the previous experiments, we noticed that RNase inhibitor and RNase A both reduced the amount of exogenous cel-miR-39. These two reagents both require a 37 °C incubation with exosomes for 1 h. We speculated if incubation at 37 °C for 1 h could cause the reduction of cel-miR-39. Here, we isolated the exosomes by ExoQuick method, and the results showed that incubation at 37 °C for 1 h reduced the amount of cel-miR-39 as much as the RNase A did (Fig. 5A). Consistent with the previous results in Figs. 3B, 3D and 4B, RNase A reduced the amount of exosomal miR-126 and miR-152, but incubation at 37 °C for 1 h reduced the amount of miR-126 but not miR-152 like the RNase inhibitor did (Fig. 5B). These results indicated that incubation at 37 °C for 1 h without RNase inhibitor

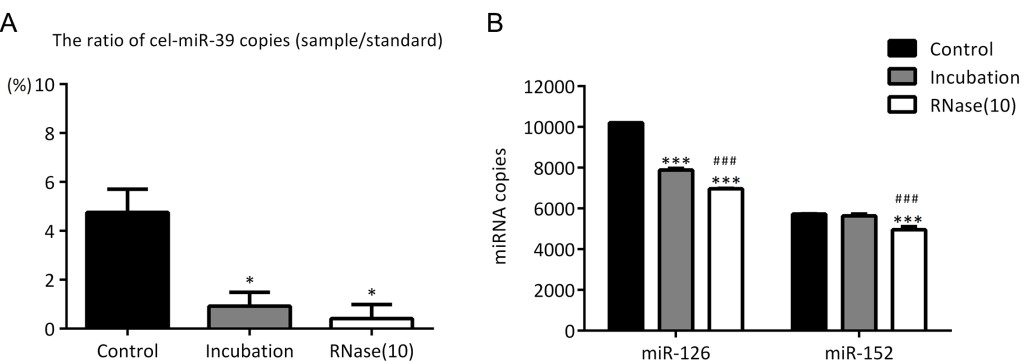

**Figure 5 Incubation at 37 °C for 1 h reduces the amount of co-isolated free miRNAs.** (A–B) A high concentration ($7 \times 10^9$ copies/μL) of cel-miR-39 was added in serum. Exosomes were extracted by the ExoQuick method. (A) The ratio of co-precipitated cel-miR-39 in different samples to standard sample ($10^9$ copies/μL cel-miR-39). *$p < 0.001$ vs. Control. (B) The amount of exosomal miR-126 and miR-152 in different samples. ***$p < 0.001$ vs. Control. ###$p < 0.05$ vs. Incubation.

and RNase A can remove the co-isolated free miRNAs from exosome pellets, but RNase A can reduce the number of miRNAs in exosomes as well.

## DISCUSSION

Circulating miRNAs, especially exosomal miRNAs, are strongly associated with the development of different diseases, such as the manifestation, invasion and metastasis of cancers (*Melo et al., 2014*; *Zhou et al., 2014*). Therefore, circulating miRNAs can serve as novel biomarkers for diagnosis, prognosis of diseases and even drugs for diseases therapy (*Cheng, 2015*; *He et al., 2015*; *Huang et al., 2015*). However, exosomal miRNAs are remarkably stable compared to circulating miRNAs that are not contained in exosomes, and different fractions of circulating miRNAs may play different roles in the progression of diseases, and serve as diagnosis biomarkers (*Grimolizzi et al., 2017*). Therefore, if miRNAs existing in exosome-free serum fractions contaminates exosomal miRNAs during laboratory studies and clinical applications, it will have a great negative impact on the sensitivity and accuracy of research and diagnostics. It has been commonly recognized that both UC and ExoQick methods co-precipitate contaminating proteins, but exosomes extracted by UC are much purer than ExoQuick on proteins (*Shao et al., 2016*; *Tkach & Thery, 2016*). However, it is still unknown if UC is also much more reliable than ExoQuick on miRNAs. Here we used exogenous cel-miR-39 as free miRNA in serum and miR-126, miR-152 as exosomal miRNAs, to investigate if and to what extent these two methods would co-precipitate free miRNAs in serum.

First, we found that compared with ExoQuick method, UC co-isolated much less exogenous cel-miR-39 with exosomes, which suggests exosomes extracted by UC are much purer than ExoQuick in exosomal miRNAs (Figs. 3A, 3C and 4A). Although 2% contaminating miRNAs may be too small to be considered, they can really make difference in some studies about exosomal miRNAs. *Rekker et al. (2014)* compared UC with ExoQuick method for miRNAs profiling, and found that 17 out of 375 miRNAs

had a slightly different levels between these two methods. Some of these 17 different miRNAs may be predominantly free in serum so that their levels in exosomes extracted by ExoQucik method were higher than that by UC. Based on our finding, when studying exosome-specific miRNAs, the standard UC method is preferred.

In previous studies, it was found that exosomal miRNAs could be protected from RNase by the bilayer membrane of exosomes and hence RNase A was used to degrade exosome-independent miRNAs (*Muller et al., 2014*; *Quackenbush et al., 2014*; *Rekker et al., 2014*). However, our results showed that although RNase A reduced the ratio of co-precipitated miRNAs (Figs. 3A, 3C and Figs. 5A), it reduced the levels of exosomal miRNAs at the same time no matter which methods were used (Figs. 3B, 3D, 4B and 5B). Furthermore, we have also found that a simple incubation at 37 °C for 1 h can reduce the amount of exosome-independent miRNAs as effectively as the addition of RNase A, but hardly reduce the amount of exosomal miRNAs at the same time (Figs. 5A–5B).

These results suggested that merely incubation at 37 ° C could remove exosome-independent miRNAs free in serum effectively, and RNase A was not recommended for degrading miRNAs out of exosomes. Earlier reports have shown that the methods of incubation at room temperature (RT) or 37 °C have been used to load drugs into exosomes (*Haney et al., 2015*; *Ma et al., 2016*). It is possible that exosomal membranes, when incubated with RNase A at 37 °C for 1 h, could display a certain level of permeability to the enzyme, and could not fully protect miRNAs from RNase A to some degree.

It is worth noting that incubation at 37 °C for 1 h (with or without RNase inhibitor) slightly reduced miR-126 instead of miR-152 (Figs. 3B, 3D, 4B, and 5B). So, there must be a difference between miR-126 and miR-152. Turchinovich et al. reported that a large part of extracellular miRNAs are associated with Ago proteins and we found from their miRNA Array data that the level of serum miR-126 associated with AGO1 and AGO2 is much higher than that of miR-152 (*Turchinovich & Burwinkel, 2012*; *Turchinovich et al., 2011*). Therefore, perhaps it is because UC and ExoQuick methods both can precipitate AGO-associated miR-126 and exosome-independent miR-126 can be degraded by incubation as cel-miR-39 did.

## CONCLUSIONS

Our study provides critical information on the effect of free miRNAs in the serum when using the UC and ExoQuick methods to isolate serum exosomes and analyze exosomal miRNAs. We have found that ExoQuick method can co-precipitate much more free miRNAs in serum than UC. Furthermore, we have also found that a simple incubation at 37 °C for 1 h can reduce the amount of exosome-independent miRNAs as effectively as the addition of RNase A, but the latter would reduce the amount of exosomal miRNAs as well. Thus, for studies of exosomal miRNA analysis, UC method is recommended, but not the use of RNase to remove exosome-independent miRNAs. In conclusion, these findings on the details of serum exosome isolation methods are essential for further studies on exosome-specific miRNAs.

### Funding

This study is supported by the National Natural Science Foundation, China (81773115 and 31801187), the SJTU Interdisciplinary Grant (YG2017MS52) and the Jiangsu Science and Technology Development Plan –Special Funds for Transformation of Science and Technology Projects (BA2011069). The funders had no role in study design, data collection and analysis, decision to publish, or preparation of the manuscript.

### Grant Disclosures

The following grant information was disclosed by the authors:
National Natural Science Foundation, China: 81773115, 31801187.
SJTU Interdisciplinary Grant:  YG2017MS52.
Jiangsu Science and Technology Development Plan –Special Funds for Transformation of Science and Technology Projects: BA2011069.

### Competing Interests

Xiangyun Qu, Zhaonan Dong, Xueqing Ma, Yunli Jia, Ruochen Li, Xiaoxu Jiang, and Tao Wang are employed by MicroDiag Biomedicine Co., Ltd.

### Author Contributions

- Yirui Cheng conceived and designed the experiments, performed the experiments, analyzed the data, prepared figures and/or tables, authored or reviewed drafts of the paper, and approved the final draft.
- Xiangyun Qu conceived and designed the experiments, analyzed the data, prepared figures and/or tables, and approved the final draft.
- Zhaonan Dong conceived and designed the experiments, performed the experiments, analyzed the data, prepared figures and/or tables, and approved the final draft.
- Qingyu Zeng analyzed the data, prepared figures and/or tables, and approved the final draft.
- Xueqing Ma, Yunli Jia, Ruochen Li and Xiaoxu Jiang performed the experiments, prepared figures and/or tables, and approved the final draft.
- Cecilia Williams, Tao Wang and Weiliang Xia conceived and designed the experiments, authored or reviewed drafts of the paper, and approved the final draft.

### Ethics

The following information was supplied relating to ethical approvals (i.e., approving body and any reference numbers):
   This study was approved by the Ethical Committee of the School of Biomedical Engineering, Shanghai Jiao Tong University (Registration No. 2019045)

### Data Availability

   The raw data are available in the Supplementary Files.

## Supplemental Information

Supplemental information for this article can be found online at http://dx.doi.org/10.7717/peerj.9434#supplemental-information.

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
