# Peer review of "Comparison of serum exosome isolation methods on co-precipitated free microRNAs"

_PeerJ, doi:10.7717/peerj.9434_

## Round 0.1 · original submission · Minor Revisions

As you will see, both expert reviewers felt your article makes an important contribution to the field, so we hope you can make these suggested changes without too much difficulty.

Reviewer 1 ·

Basic reporting

The manuscript is well written and clearly and logically set out. It cites a range of relevant literature. The figures are presented logically with detailed legends and the results are all relevant to the aims of the manuscripts. The materials and methods are detailed and easy to follow. Raw data is provided for the experiments presented in the paper.

Experimental design

The field of exosome biology and their microRNA content and role in health and disease is growing rapidly. There are many different approaches to isolating and characterising exosomes and their contents. Although clear guidelines are being constantly updated regarding appropriate QC for exosome work there are few papers published that directly compare different methods for exosome isolation and determine the amount of free microRNA carried through purification. This is important knowledge and will help researchers refine their studies and ensure that conclusions are robust.
The ethics declaration form is in Mandarin with the exception of reference to the mouse strain "Balb-c". An English language version should be provided which has relevant ethics for the studies presented in the manuscript.

Validity of the findings

The results are clearly presented in a logical manner. It is usual to present more than one exosome/ EV marker by western blot. The authors should add in additional western markers as quality controls for their isolated EVs, e.g CD63, or another of the accepted markers. Furthermore, the TEM images, particle size distribution and western images in Figure 1 are not very clear. The authors should adjust the size/ resolution of these images so that they are clearer for the reader.The discussion and conclusions are valid and clear.

Additional comments

This is a useful paper which should be informative to researchers working in the field of exosome/ EV biology and who wish to ensure the methods they use for purification are appropriate and that their results and conclusions are robust.

Reviewer 2 ·

Basic reporting

The paper by Cheng et al. reports on the comparison of serum exosome isolation methods, including ultracentrifugation and precipitation-based commercial kit, on co-precipitated free microRNAs. The subject of this study is topical and relevant for the expanding field of exosome isolation since new methods currently are commercially available with considerable benefits in terms of exosomal purity, amount and labor consumption. Thus, in this work, the authors have attempted to validate methodologies that will help better providing details of serum exosome isolation methods for further research on exosomal miRNAs.
Overall, the topic is of interest and the work carefully executed. The paper is well written.

Experimental design

Some reservations regarding the conclusion and the methodology:


The miR126 and miR152 copies of the blank control sample in the different figures as well as the control sample are not stable.

Please, specify why has not any housekeeping gene been used as an endogenous control for the quantification of the number of copies of miR126 and miR152.

Thus, I encourage the authors to discuss and clarify this point to prevent misleading potential readers

Validity of the findings

-is the exosome characterization by size using Nanosight a single experiment? Please, specify in the legend or add the other biological replicates

-please, add the concentration of miR-39 in the figure 2, even if it is included in the legend, to be clearer

-figure titles and legends, please add spaces between words and add/remove spaces into the brackets

-please, add the molecular weight-size in the western blots in the supplementary figures

The data are robust and statistically sound

Additional comments

Overall, this is an interesting paper which should help the scientific community better to improve the methods developed for exosome isolation eliminating serum contaminants

---

## Round 0.2 · accepted · Accept

Thanks for attending to these minor revisions. I look forward to seeing your work in print!